# Association of Self-Reported Depression Symptoms with Physical Activity Levels in Czechia

**DOI:** 10.3390/ijerph192114319

**Published:** 2022-11-02

**Authors:** Geraldo A. Maranhao Neto, Eduardo Lattari, Bruno Ribeiro Ramalho Oliveira, Anna Bartoskova Polcrova, Maria M. Infante-Garcia, Sarka Kunzova, Gorazd B. Stokin, Juan P. Gonzalez-Rivas

**Affiliations:** 1International Clinical Research Center (ICRC), St. Anne’s University Hospital (FNUSA), 602 00 Brno, Czech Republic; 2Postgraduate Program in Physical Activity Sciences (PGCAF), Salgado de Oliveira University, Niterói 24030-060, Brazil; 3Department of Physical, Education and Sports, Physical Activity, Health, and Performance Research Laboratory, Rural Federal University of Rio de Janeiro, Seropédica 23890-000, Brazil; 4Research Centre for Toxic Compounds in the Environment (RECETOX), Masaryk University, 625 00 Brno, Czech Republic; 5Foundation for Clinic, Public Health, and Epidemiology Research of Venezuela (FISPEVEN INC), Caracas 3001, Venezuela; 6Department of Global Health and Population, Harvard TH Chan School of Public Health, Harvard University, Boston, MA 02138, USA

**Keywords:** mental health, depression, physical activity, population health, adult, middle age

## Abstract

Worldwide, depressive disorder is one of the leading determinants of disability-adjusted life years. Although there are benefits associated with a higher physical activity (PA) level, there is a lack of information related to this relationship, especially in countries such as Czechia, where modern approaches to mental health care only recently emerged. The present study aimed to evaluate the association between the level of depression and different PA levels following the World Health Organization (WHO) PA guidelines and according to specific symptoms that indicate depression. Multivariable-adjusted Poisson regression models were used to calculate the prevalence rate (PR) in a sample of 2123 participants (45.3% men, median 48 years). Compared to subjects with insufficient PA, moderate and high PA levels were inversely associated with continuous depression scores (PR = 0.85; 95% CI: 0.75–0.97; and PR = 0.80; 95% CI: 0.70–0.92). Depressed mood and worthlessness were the symptoms associated with moderate and high PA. Tiredness, change in appetite, and concentration problems were related to high PA. The results suggest that reaching the minimum PA target according to the guidelines seems to be effective, and this could stimulate adherence. However, more specific improvements in symptomatology will require a subsequent gradual increase in PA levels.

## 1. Introduction

Major depressive disorder is defined as the experience of a depressed mood for at least two weeks or losing pleasure in most activities, which is not caused by a temporary medicine or medical condition [1]. Worldwide, depressive disorder is one of the leading determinants of disability-adjusted life years (DALYs) [2], affecting over 300 million subjects in 2015 (4.4%) [3]. In Europe, about 53 million adults suffer from depression, and Central and Eastern European countries (CEE) report higher prevalences than other parts of Europe [3]. CEE has experienced profound structural changes since the fall of communist regimes in 1989 [4], when health promotion and primary prevention concepts first emerged in countries such as Czechia [5]. Since then, numerous policy initiatives have emerged to improve care for people with mental illness. However, modern approaches to mental health care have emerged only recently [6], intending to provide mental health services that are achievable, of high quality, and based on needs and providing programs for promotion, prevention, and mental health education [7]. Despite this, there is still a lack of sufficient information on the prevention of mental disorders [8]. 

The complexity of depression presents considerable challenges to traditional treatments, which include pharmacotherapy, psychotherapy, or a combination of both. These forms of therapy provide valuable benefits for treating depression [9]. However, 30% of patients do not respond to the current pharmacotherapy, and 70% do not achieve complete remission [10]. In addition, the undesirable side effects and the high economic and social costs to society have diversified the search for more cost-effective options. Of the current alternatives, physical activity has been a consistent adjunct in the treatment and prevention of depressive symptoms [11].

Physical activity (PA) represents all body movements in everyday life that lead to the expenditure of energy above the resting state [12]. Although the mechanisms by which exercise modulates depression remain unclear, several studies have reported that higher levels of PA contribute favorably to the prevention and treatment of depression [9,13,14]. While Czechia is considered an “active country” [15] of “walkers and cyclists” [14], the prevalence of physically inactive adults has increased from 37.3% in 2013 to 42.7% in 2017 [16]. Among 36 countries, Czechia had the highest and most significant association between the presence of depression and low levels of PA (odds ratio: 6.02) [17], and evidence shows that this association has been detected earlier in Czech adolescents [18]. 

Recently, new World Health Organization guidelines on physical activity and sedentary behavior provided evidence-based public health recommendations concerning the amount and types of physical activity that offer significant health benefits and mitigate health risks [19]. However, there is still a low number of studies evaluating the comparison between moderate (minimum recommendation) and a high level of PA (moderate–vigorous) [20,21]. Furthermore, the association of PA with symptoms of depression is usually reported generically, without detailing specific symptoms, especially considering the heterogeneity of these manifestations (cognitive, emotional, and somatic) [22,23]. In addition, there does not seem to be any population-based study that has verified the association of specific symptoms of depression with the level of PA while controlling for possible confounders. Thus, the hypothesis was that individuals who reach the minimum PA recommendation would present an inverse association with the level of depression and that this association would increase with a higher level of PA. However, this relationship would differ according to specific depression-related symptoms.

Therefore, this study aimed to evaluate, in a probability-based sample of the Czech population, the association between the level of depression and different PA levels as well as the association according to each specific symptom that characterizes depression.

## 2. Materials and Methods

### 2.1. Study Design and Population

The Kardiovize study is a cross-sectional population-based study with a random sample of 1% of the adult (25–64 years old) population of Brno, Czech Republic. Brno, the second largest city in the Czech Republic, had 373,327 residents in 2013. The eligibility criteria included permanent residence in Brno and registration with any of the five state-run health insurance companies operating in the country, covering 91.1% of the population.

### 2.2. Sampling

A random sample of 3300 persons that was stratified by age and gender was adjusted for a response rate of 64.4%, as projected from the Czech post-MONICA study. The MONICA (Multinational MONItoring of Trends and Determinants in CArdiovascular Disease) project, for a period of ten years or more from the early 1980s to the mid-1990s, implemented cardiovascular disease surveillance in 21 countries. It included mortality, morbidity, coronary care, and population-based risk factor surveillance [24]. In 1997/1998, 2000/2001, 2006–2009, and 2015–2018, Czechia followed up on the MONICA study and conducted four additional independent cross-sectional surveys of a randomly selected 1% population sample of individuals aged 25–64 years in nine counties (Czech post-MONICA study) [25]. 

Health insurance companies mailed invitation letters with a description of the study, ensuring confidentiality. Because the sample size was not reached, a second random sample was obtained following the same methodology as the first sample. For the second invitation, 3077 invitations were mailed. Based on the two samplings with a total of 6377 randomly selected invitees, the overall response rate was 33.9%. No information on non-respondents was available. A total of 2160 individuals signed informed consent to participate and were enrolled in the Kardiovize study. The Kardiovize study was approved by the ethics committee of St Anne′s University Hospital, Brno, Czech Republic (ref. number: 2G/2012).

### 2.3. Data Collection

A questionnaire was developed, which included information about demographics (e.g., age, education, and marital status), socioeconomic status, and cardiovascular risk behaviors (e.g., smoking and alcohol consumption). Laboratory analyses were performed on 12 h fasting whole blood samples using a Modular SWA P800 analyzer (Roche, Basel, Switzerland). Glucose was analyzed by the enzymatic colorimetric method (Roche Diagnostics GmbH, Mannheim, Germany). Blood pressure was measured with the patient alone using an automated office measurement device (BpTRU, model BPM 200; Bp TRU Medical Devices Ltd., Coquitlam, BC, Canada). Body composition analyses were performed using a scale with bioelectrical impedance analysis capabilities (InBody 370; BIOSPACE Co., Ltd., Seoul, Korea) [25]. The information regarding each individual was obtained within one week.

### 2.4. Variable Definitions

Hypertension was defined as BP ≥ 140/90 mmHg, self-reported hypertension, or the use of anti-hypertensive medications. Diabetes was defined as fasting blood glucose ≥7 mmol/L, a self-report of diabetes, or taking antidiabetic medications. The cutoff points for high body fat were 25% for men and 35% for women. Self-reported PA was assessed using the International Physical Activity Questionnaire (IPAQ) long version [25] with a historical record according to the last week. Subjects classified as “high PA” were those who participated in a vigorous-intensity activity at least 3 days per week, achieving a minimum of 1500 MET-minutes/week or 7 days per week with any combination of walking, moderate-intensity activities, or vigorous-intensity activities achieving a minimum of 3000 MET-minutes/week. Subjects classified as “moderate PA” were those who participated in at least 20 min of vigorous physical activity 3 or more days per week, or at least 30 min of moderate-intensity physical activity or brisk walking 5 or more days per week, or 5 or more days per week of any combination of walking, moderate-intensity activities, or vigorous-intensity activities, achieving a minimum of 600 MET-min/week. Subjects classified as “insufficient PA” were those who did not reach the activity levels listed above. The weekly MET minutes were calculated by multiplying the MET factor assigned to each activity (walking = 3.3 MET, moderate-intensity activity = 4.0 MET, vigorous-intensity activity = 8.0 MET) by the duration (in minutes) and the number of days that the respective activity was performed [26]. 

Marital status was classified as “living alone” (including single, divorced, and widowed) or “living in a couple” (including married and other partnerships). Educational level was classified as primary, secondary, and higher (representing the highest level achieved). Household income was expressed in EUR per month and classified as “low” (<1200), “middle” (1200–1800), or “high” (>1800). Smoking status was classified as “Non-smokers” or “current smokers” (smoking in any amount during the past year). Alcohol consumption was assessed by the reported alcohol intake of the last week, expressed in the number of standard drinks. One standard drink contained approximately 10 g of ethanol (100–125 mL of wine, 250 mL of beer, or 30 mL of spirits [27]. Participants were classified as “non-alcohol users” (including abstainers and those who did not drink in the previous 12 months) and “alcohol users” [25].

### 2.5. Outcomes of Depression

Depressive symptoms were assessed by the Patient Health Questionnaire (PHQ-9), a nine-item questionnaire designed to screen for depression [28]. The PHQ-9 consists of nine questions that assess the presence of each of the symptoms of an episode of depression described in the Diagnostic and Statistical Manual of Mental Disorders (DSM-IV). The nine symptoms consist of depressed mood, anhedonia (loss of interest or pleasure in doing things), problems with sleep, tiredness or lack of energy, change in appetite or weight, feelings of guilt or worthlessness, problems with concentration, feeling slow or restless, and suicidal thoughts. The frequency of each symptom in the last two weeks is evaluated on a 4-point Likert scale from 0 to 3, corresponding to the answers “Not at all”, “Several days”, “More than half the days”, and “Nearly every day”, respectively. PHQ-9 final scores range from 0 to 27.

### 2.6. Data Analysis

Analyses were performed with STATA software (version 14.0, StataCorp, College Station, TX, USA). The Kolmogorov–Smirnov test was used to assess the normal distribution of variables. Continuous variables were reported as medians and ranges and were compared using the Mann–Whitney U test. Categorical variables were reported as percentages and compared using the chi-squared or Fisher tests. The depression outcomes were determined in two different approaches: (1) the total score from the PHQ-9 (0–27) and (2) each symptom of the PHQ-9 separately (0–3). A Poisson regression model was used to test for the categories of each symptom as counts or frequencies. For the total PHQ-9 score, a zero-inflated Poisson regression was used to model count data with an excess of zero counts. For all analyses, a generalized Poisson model using robust error variances was applied to estimate appropriately narrow 95% confidence intervals (CI) of the raw and adjusted prevalence rates (PR). The variables included in the full model were age [29], gender [30], educational level [31], household income [32], living in couple [32], alcohol use [33], high body fat percentage [34], and diabetes [35]. Age and gender were included in model 2, and all variables were included in model 3.

## 3. Results

### 3.1. Subject Characteristics

In total, 2123 subjects that answered the questions about depression were included (45.3% men, with a median age of 48.0 (IQR 19.0) years). The classification of PA level was: insufficient (13.8%), moderate (36.2%), and high (50%). The median PHQ-9 scores were 2 points in males and 3 points in females (*p* < 0.001). The score was also higher in subjects between 25 and 44 years old and in those with primary education, with a low household income, not living in a couple, with a lack of alcohol consumption, and with a high body fat percentage (Table 1). Figure 1 highlights the difference between insufficient and high PA levels. 

### 3.2. Adjusted Association between Physical Activity Level and Depression

In the raw model, moderate PA had an inverse association with significantly lower values (from 15% to 18%) in PR, suggesting the importance of moderate PA in influencing small changes in depression levels, even though it was not enough to change the depression categories. A high PA level was also significantly associated with the depression score (PR decreasing from 21% to 23%) (Table 2).

### 3.3. Association between Physical Activity Level and PHQ-9 Symptoms

Compared with the low level of PA, “feeling down”, “depressed”, or “hopeless” was 34% and 41% lower in the moderate and high PA levels, respectively, and “feeling bad about yourself” was 26% and 34% lower in those with moderate and high levels of PA, respectively (Table 3). In turn, only those with high PA levels had significantly lower PRs of “feeling tired” (24%), “poor appetite or overeating” (25%), and “trouble concentrating” (29%). This was not seen in those with moderate levels of PA (Table 3).

## 4. Discussion

Czech subjects who engaged in moderate PA (achieving the minimum recommendation by the WHO—at least 150–300 min of moderate-intensity aerobic physical activity) presented a 15% lower proportion of scores of depression. When engaging in high PA (at least 75–150 min of vigorous-intensity PA), a 20% lower proportion was observed. There were some particularities of the present study that are different from the literature. For instance, the study by Asztalos et al. [36] in a population-based sample of 3368 men and 3435 women aged 25–64 years from the Belgian Health Interview Survey did not present the category “insufficient” or “inactive” to compare. In the general sample, the associations between PA and mental health indices were always positive and not inverse, regardless of the PA intensity. Currier et al. [21], when analyzing 13,763 men in the Australian Longitudinal Study on Male Health, found 40% lower odds of having depression at the moderate PA level (odds ratio = 0.60; 95% CI: 0.53–0.68). However, there was no comparison to the category “insufficient”. Moreover, Marques et al. [20] showed a cross-sectional and prospective relationship between physical activity and depression. Moderate and vigorous PA at least once a week were negatively related to the score of depression. This supports the idea that some physical activity is better than none, but a gradual increase would bring much more significant benefits, not only for mental health, and should be emphasized. It is noteworthy that a limitation of the study was not including any adiposity indices. Both overall adiposity (total body fat and BMI) and abdominal adiposity (waist circumference and visceral adipose) measures were associated with depressive symptoms [34]. In a recent meta-analysis, Pearce et al. [37] showed the first dose–response meta-analysis between physical activity and incident depression, with a 25% lower risk of depression (0.75; 95% CI: 0.68–0.82) for moderate PA and a 28% lower risk for vigorous activities (0.72; 95% CI: 0.64–0.81). Despite only being used in cross-sectional studies, the prevalence rate of the present study is mathematically identical to the relative risk calculation, with comparable results. 

Most analyses have investigated depression as a syndrome without considering specific symptoms that may decrease the chance of engaging in physical activities [21]. In the present study, moderate and high levels of PA were related to some symptoms of depression. The current findings suggest that high levels of PA seem to contribute to a lower prevalence of 5 of 9 specific symptoms of depression. There was no association with anhedonia, sleep problems, feelings of slowness or restlessness, or suicidal thoughts. Comparisons with other studies are limited, as most of them include preselected individuals with symptoms of depression [38,39]. Nevertheless, similar results can be observed in the study by McKercher et al. [39], which was carried out with 1995 individuals (31.5 ± 2.6 years) with major depression. Depressed mood was inversely associated with moderate and high PA. Mood is probably the most studied symptom related to depression [40]. Hollands et al. [40], in 43 preselected participants with depression, suggested that mood appears to be affected by any level of physical activity. Feelings of guilt or worthlessness were also inversely associated by moderate and high PA. McKercher et al. [39] showed an inverse association with leisure PA in young men and women with major depression but without discriminating the PA level. Most studies have only considered the presence/absence of PA without a discrimination of the volume or intensity. Three specific symptoms (tiredness, poor appetite or overeating, and trouble concentrating) were inversely associated only in individuals who reported high levels of PA. The study that may have the most similar characteristics to compare is that of Althumiri et al. [41] in a non-clinical population (n = 8333) from Saudi Arabia. Moderate PA and vigorous PA were significantly associated with fewer depressive symptoms for six and three of the nine items, respectively. However, the statistical analysis did not adjust for any covariables and essentially compared each symptom score in the different PA levels by t-test, which seemed to be an inappropriate approach.

Some mechanisms can help to understand the symptoms associated with increased PA. Regular PA can regulate neuronal plasticity through the expression and activation of brain-derived neurotrophic factor (BDNF) and 5HT release [42]. Furthermore, at higher levels of PA, the release of beta-endorphins from the pituitary modulates hedonic and appetite-motivated behaviors [42]. There is also evidence suggesting that as individuals increase PA they also increase the frequency of feelings of energy and decrease the frequency of feelings of fatigue [43]. The specific mechanisms associated with increased energy and decreased fatigue are not fully understood. However, monoamine, histamine, acetylcholine, glutamate, and gamma-aminobutyric acid (GABA) are involved. In addition, physical activity could regulate these neurotransmitters [44]. Regular PA can promote adrenal hormone receptor activity and improve the production and secretion of dopamine and norepinephrine, thereby improving concentration and working memory [38]. In addition, specific brain regions, such as the prefrontal and parietal cortices, benefit most from aerobic physical activity [45]. High-intensity exercise improves brain indices, reflecting executive and sustained attention during task performance [46]. The PA is thought to have its antidepressant effect through multiple biological and psychosocial pathways, and it could influence depression in two ways: with a preventive value (it is used to protect against the development of depressive symptoms) and as a “treatment” [47]. Changes occur in the brain to produce an environment that is protective against depression. For instance, cellular processes such as angiogenesis are stimulated by neurotrophins, causing changes in brain blood flow that improve functioning [9]. Moreover, psychosocial factors accompany and interact with these biological changes to influence depression, such as the distraction of stressful stimuli, better self-esteem, greater control over your body and your life, and social interaction [48]. 

The associations between PA levels and depression have also been dependent on different outcomes that are used to estimate the symptoms of depression. The advantage of the score is the detection of minor contrasts in the level of depression. The importance of using scale responses continuously has been discussed in the literature [37]. However, presenting results in terms of response versus non-response or with different categories gives the impression that the effect of physical activity on the symptoms of depression is a “presence/absence” phenomenon [49]. In recent years, the association between PA level and the risk of depression has been investigated, suggesting an inverse relationship. More studies have analyzed different intensities of PA and the risk of depression. However, the results should provide information from a public health perspective that makes it possible for the population to understand that they can adapt to the current guidelines. This would be one of the gaps filled by the present study.

The Czech National Action Plan for Mental Health (NMHAP) 2020–2030 is a document that considers that the concept of a biopsychosocial model is increasingly well established [50]. In this context, it is not enough only to provide quality care to people with mental illness. It is necessary to focus on maintaining the mental health of the entire population, adding to the current approach the aspect of prevention and early intervention. In this sense, the role of PA interventions in promoting mental health has evolved to be considered an effective, safe, inexpensive, and widely accessible preventive and therapeutic modality [51].

Some limitations should be considered in the interpretation of this study. The IPAQ questionnaire is not specific concerning the exercise type, and only PA performed continuously for at least 10 min is considered [52]. Therefore, activities with interval characteristics such as resistance exercise may not be appropriately recorded, despite their benefits for depression [53]. On the other hand, the use of questionnaires for PA estimation is necessary for epidemiological studies due to the large sample sizes. No control for depression medication was considered. Thus, people with depression that were being treated with medications could have been included in the sample and influenced the results. Moreover, the PHQ-9 does not replace a clinical diagnosis, but it has a sensitivity that is substantially greater than the semistructured reference standards [54]. The cross-sectional design is another limitation because it does not allow a causal relationship, and the relationship between PA and depression is bidirectional. However, the study has strengths that must be highlighted: it is the first investigation that demonstrates specific associations according to the most recent guidelines suggested by the WHO, and specifically it evaluated the associations of particular symptoms related to depression and did not only treat the outcomes of depression as a syndrome. In addition, there was an adjustment for a series of covariables, including the percentage of body fat, a more accurate indicator than the BMI alone, which is more commonly used in epidemiological studies. 

## 5. Conclusions

In conclusion, the present study showed that those achieving the minimum PA recommendation (moderate PA) had a 15% lower level of depression, which increased to 20% with a higher PA. However, regarding specific symptoms, only depressed mood and feelings of guilt were associated with moderate and high PA. Three specific symptoms (tiredness, lack of appetite or overeating, and difficulty concentrating) were only inversely associated with higher PA. From the perspective of public health and the promotion of physical activity, the results suggest that reaching the minimum PA target according to the WHO guidelines seems to be effective, and this could stimulate adherence. However, more specific improvements in symptomatology will require a subsequent gradual increase in PA levels.

## Figures and Tables

**Figure 1 ijerph-19-14319-f001:**
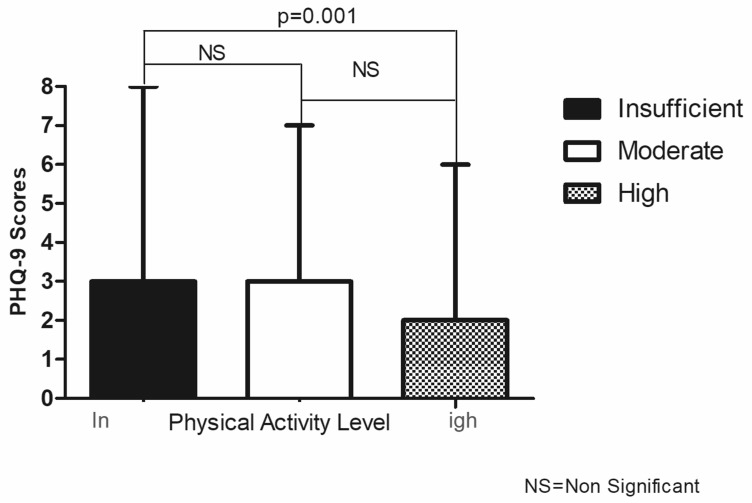
Patient Health Questionnaire-9 (PHQ-9) Scores (Median, Interquartile) and Physical Activity Level.

**Table 1 ijerph-19-14319-t001:** Population characteristics across PHQ-9 scores.

	PHQ-9 (Score Median Range)	*p*
Gender		
Male	2.0 (0–20)	
Female	3.0 (0–23)	<0.001
Age Categories		
25–34	3.0 (0–20)	
35–44	3.0 (0–19)	
45–54	2.5 (0–23)	
55–64	2.0 (0–23)	0.023
Educational Level		
Primary	3.0 (0–23)	
Secondary	2.0 (0–23)	
Higher	2.0 (0–20)	0.002
Household income (EUR)		
Low (<1200)	3.0 (0–23)	
Middle (1200–1800)	2.0 (0–18)	
High (>1800)	2.0 (0–19)	<0.001
Living in Couple		
No	3.0 (0–23)	
Yes	2.0 (0–19)	0.002
Alcohol Users		
No	3.0 (0–18)	
Yes	2.0 (0–23)	0.006
Smokers		
No	2.0 (0–23)	
Yes	2.0 (0–23)	0.591
High Body Fat Percentage		
Absent	2.0 (0–23)	
Present	3.0 (0–20)	0.002
Diabetes		
Absent	2.0 (0–23)	
Present	3.0 (0–14)	0.370
Hypertension		
Absent	2.0 (0–23)	
Present	2.0 (0–23)	0.443
Cardiovascular Disease History		
Absent	2.0 (0–23)	
Present	3.0 (0–17)	0.312
The Mann–Whitney U test was used to determine different medians.		

**Table 2 ijerph-19-14319-t002:** Association between physical activity level and depression outcomes.

	Physical Activity Level # and PHQ-9 Scores		
Physical Activity Level	Model 1	Model 2	Model 3
	PR (95% CI)	PR (95% CI)	PR (95% CI)
Insufficient	1	1	1
Moderate	0.85 * (0.74–0.97)	0.81 ** (0.71–0.93)	0.85 * (0.75–0.97)
High	0.80 ** (0.70–0.91)	0.78 *** (0.68–0.88)	0.80 ** (0.70–0.92)

# Low level as the reference; * < 0.05; ** < 0.01; *** < 0.001; Model 1—raw model; Model 2—adjusted by age categories and gender; Model 3—adjusted by age categories, gender, educational level, household income, living in couple, alcohol users, high body fat percentage, and diabetes; PR—prevalence rate; CI—confidence interval.

**Table 3 ijerph-19-14319-t003:** Association between physical activity level and PHQ-9 items.

		Physical Activity Level #	
		Moderate	High
		PR (95% CI)	PR (95% CI)
“Little interest or pleasure in doing things”		0.90 (0.73–1.12)	0.83 (0.67–1.06)
“Feeling down, depressed, or hopeless”		**0.66 (0.48–0.91)**	**0.59 (0.44–0.81)**
“Trouble falling or staying asleep, or sleeping too much”		0.96 (0.80–1.16)	0.97 (0.81–1.16)
“Feeling tired or having little energy”		0.89 (0.78–1.08)	**0.86 (0.76–0.98)**
“Poor appetite or overeating”		0.80 (0.61–1.06)	**0.75 (0.57–0.98)**
“Feeling bad about yourself or that you are a failure or have let yourself or your family down”		**0.74 (0.56–0.98)**	**0.66 (0.51–0.86)**
“Trouble concentrating on things, such as reading the newspaper or watching television”		0.84 (0.64–1.10)	**0.71 (0.54–0.94)**
Moving or speaking so slowly that other people could have noticed. Or the opposite, being so fidgety or restless that you have been moving around a lot more than usual		0.68 (0.43–1.08)	0.69 (0.44–1.08)
“Thoughts that you would be better off dead, or of hurting yourself”		0.96 (0.36–2.51)	0.64 (0.24–1.70)

# Low level as the reference; PHQ—Patient Health Questionnaire; PR—prevalence rate; CI—confidence interval.

## Data Availability

The data presented in this study are available upon request from the corresponding author. The data are not publicly available.

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
