# Peer review of "Association of Self-Reported Depression Symptoms with Physical Activity Levels in Czechia"

_ijerph, 2022, doi:10.3390/ijerph192114319_

Round 1

Reviewer 1 Report (Previous Reviewer 1)

This is a cross-sectional study report with the title “Association of Self-Reported Depression Symptoms with Physical Activity Levels in Czechia”. There is no new clinical implication for the readers. The reviewer has the following comments about this manuscript.

 1. The authors did not respond to the questions point by point. Clinical implication is still unclear with ambiguous PA status and ambiguous depressive status of participants in this manuscript, especially without clarification of clinically diagnosed depression.

Error “4. Friborg, O.; Rosenvinge, J. H., A comparison of open-ended and closed questions in the prediction of mental health. Quality & Quantity 2011, 47, (3), 1397-1411.” should be changed to “4. Friborg, O.; Rosenvinge, J. H., A comparison of open-ended and closed questions in the prediction of mental health. Quality & Quantity 2013, 47, (3), 1397-1411.”

2. The authors again did not respond to the questions point by point, such as “Did they receive medical treatment or have any physical comorbidity?” The definitions of the variables for hypertension and diabetes were too vague without clinical diagnoses across all samples, such as allowing for self-reported diagnosis. For example, one data point of blood pressure above or equal to 140/90 mmHg does not mean clinical diagnosis of hypertension. Furthermore, there may be conflicts in definition if a participant was taking anti-hypertensive or antidiabetic medication, but was below BP 140/90mmHg or fasting blood glucose 7mmol/L.

3. The authors did not respond clearly to the questions point by point. What does “the same period of time” refer to? As this is a cross sectional study, it will be hard to determine the association of PHQ-9 score and PA with unclear variables.

4. The authors declare that “The cross-sectional design is another limitation because it does not allow a causal relationship, and the relationship between PA and depression is bidirectional”. That is correct!

5. That is correct that PHQ-9 can be sensitive for possible depressive symptoms, but does not replace clinically diagnosed depression.

6. However, there is still no novelty in the conclusion.

Author Response

Dear Reviewer,

the answers are attached.

Reviewer 2 Report (Previous Reviewer 3)

The article  Association of Self-Reported Depression Symptoms with Physical Activity Levels in  Czechia focuses on searching for connections between depressive disorder and undertaking various forms of physical activity in adults. In face of increasing number of depressive patients around the world, the researched issue is socially and clinically very important. The aim of the study was to evaluate the association between PA levels and different aspects of depression in Czech population. The presented study called the Kardiovize study is a cross- sectional, randomized evaluation of adults between 25-64. The participants were 2 123 inhabitants  of Brno, in  Czech Republic. The used measurement tools were : the Patient Health Questionnaire (PHQ-9),( Depressive symptoms) and the International Questionnaire of Physical Activity (IPAQ) long version (Self-reported PA).

Demographic data were gathered and laboratory analyses were performed .
As statistical method the multivariable-adjusted Poisson regression models were carried out .
The manuscript is generally clear, relevant for the field and presented in a well-structured manner.
Cited references are current . Ethical standards preserved.

The current version of the paper is completed according to reviewer’s suggestions.
New information relate two projects mentioned in the first paper version Kardiovize-study  and Czech post-MONIKA study.
The projects were described  previously not enough, what would cause unclarity in readers.
Current supplement explains the research design, what is very helpful.
The strengths of the paper:

1.     The project has an  interdisciplinary character, with the strong  social, psychological and medical  background.

2.     The findings are of importance: the present study showed that PA levels have an inverse relationship with depressive symptoms, minimum a moderate level of PA is recommended for people with self-reported depression.

3.     The research results are evidence-based directions for public health policies .

4.     Deep discussion, which presents rich knowledge, good interpretation and synthesis.

The weaknesses of the paper: The current completed version of the  paper seems to be good prepared and deeply considered. Previous mistakes have been removed.

Author Response

Thank you for the comments.

Reviewer 3 Report (Previous Reviewer 4)

INTRODUCTION

I find it clear, well written and interesting.

It is missing the hypotheses of the study; I suggest including it together with the objective of the study (line 70).

Answer to revision: I suggest if it is possible to specify the hypothesis. For example, mention if there is any aspect or symptom of depression that will show a stronger association with the level of physical activity.

MATERIALS AND METHOD

Adequate description of materials.

Accurate methodological design.

RESULTS

I suggest adding graphs to complement the presentation of the results.

Answer to revision: Add standard error bars in figure 1.

DISCUSSION

I consider it is appropriate and well written. It highlights the main findings of the study.

The paper titled “Association of Self-Reported Depression Symptoms with Phys-ical Activity Levels in Czechia” addresses an interesting topic that provides evidence about the benefit of physical activity in the emotional state. In my opinion, these data are relevant for the promotion and prevention of mental health.

Author Response

Dear Reviewer,

the answers are attached.

Reviewer 4 Report (New Reviewer)

I enjoyed reading the article. The article gives prominence to the Czech Republic as a country that moves between history and the past, and the Czech Republic as a country that corresponds with the values ​​of the Western world.

In this context, one of the things I was missing both in the introduction and in the discussion chapter is a theoretical model that would organize the examined variables within their context. You can choose an ecological model that will refer to the participants' historical, social and personal circles, alternatively a model that will talk about self-regulation in the context of sports activities.

In addition, it is important to address the question of whether the participants received compensation for their participation, given the fact that a commercial company was involved. Finally, I would also like to see in the discussion chapter a reference to the understanding of the findings in the cultural-social context of the Czech Republic.

Author Response

Dear Reviewer, the answers are attached.

This manuscript is a resubmission of an earlier submission. The following is a list of the peer review reports and author responses from that submission.

Round 1

Reviewer 1 Report

This is a cross sectional study report with the title “Association of Self-Reported Depression Symptoms with Physical Activity Levels in Czechia”. The authors were motivated to evaluate the association between PA levels and different aspects of depression such as:
a) clinician diagnosed depression history based on patient’s history;
b) different severities of depression according to a questionnaire;
c) continuous depression score from a questionnaire; and
d) the association of different symptoms that characterize the depression.  There is no new clinical implication for the readers. The reviewer has the following main questions about this manuscript.

1.      The definition of “clinician diagnosed depression history” is inadequate to meet the expectations of clinicians. The clinician-diagnosed depression history [20] was assessed through the question: “Have you ever been diagnosed with depression?” However, the cited reference 20 mentions their definition “participants were asked whether they have been diagnosed with depression by a physician or psychotherapist (last 12-month)”. How long were the participants treated under a clinician’s service? Are these participants at the status of depressive episode, partial remission or full remission? Clinical implication is unclear with ambiguous PA status and depressive status of participants in this manuscript.

2.      In section 2. Materials and Methods & 2.1. Study Design and Population on page 2, the authors mentioned “The study design was described previously [31]”. However, the reviewer did not find any study designs in the reference 31: Alcohol Use Disorder and Depressive Disorders. In this section, the authors mentioned laboratory analyses by checking 12-hour fasting whole blood, physical examination by checking blood pressure, and body composition analyses. However, the reviewer still questioned how the participants were diagnosed as diabetes, hypertension, and high body fat percentage. Did the participants get the diagnosis of diabetes, hypertension, and obesity by a clinician? Did they receive medical treatment or have any physical comorbidity?  

3.      Self-reported PA was assessed using the International Questionnaire of Physical Activity (IPAQ) long version [24]. Although the cited reference revealed “usual week” and “last 7 d” reference periods performed similarly, and the reliability of telephone administration was similar to the self-administered mode, the reference also recommended the long form for research requiring more detailed assessment in their conclusion. How did the author define the time point of checking IPAQ and its relationship with depression score? Did the participants check their PHQ-9 score repeatedly after they performed moderate PA or high PA? Or did the participants just evaluate their PHQ-9 score occasionally under their usual moderate PA or high PA?

4.      It is well known that being physically active allow you to increase your fitness level, improve your bone health, reduce the risk of suffering from many different conditions, such as hypertension, disease, diabetes and even cancer, reduce the risk of falls, ensure that you keep a healthy weight, and improve your mood. It is also well known that depressive patients have decreased physical activity, with defined depression symptoms such as lack of energy/interest, fatigue, and psychomotor retardation. It results an inverse relationship between moderate and high PA levels with clinician-diagnosed depression history and depression scores. The self-reported depression symptoms and physical activity will mutually affect each other in a vicious cycle. This study design could not clarify that higher levels of PA contribute favourably to the prevention and treatment of depression. As a cross-sectional survey, it is difficult to differentiate the causes and effects of household income, living as a couple, alcohol use, high body fat percentage and diabetes. It is therefore difficult to determine the relationship between self-reported depression symptoms and physical activity. To confirm the effects or association of these factors, a cohort study should be designed. It is difficult to make further conclusions or novelty knowledge based on a cross-sectional survey.

5.      As the author stated in the manuscript, “Depressive symptoms were assessed by the Patient Health Questionnaire (PHQ-9), a clinical screen for depression [26]”. The PHQ-9 is only a screening tool for depression and is not equivalent to a professional diagnosis. Thus, it has no important significance for clinical implication because the participants’ condition at that time might not have clinically significant impaired function such as decreased self-care or social interaction.

6.      The depression symptoms outcome is still ambiguous. The conclusion that high PA was inversely associated with less self-reported depression symptoms, and that many factors (such as low income, alcohol use, living in couple, diabetes, obesity) would confound the association of moderate PA and self-reported depression symptoms is common knowledge. There is no novelty in this hypothesis to be tested in these participants.

 Page 6 Table 6 line 12 “secundary” is mistake of “secondary”.

Reviewer 2 Report

1.  The study design section should include how the experiment was carried out with the overall study procedure. 

2. Data collection section is irrelevant. Blood sampling, blood pressure measurement, and body composition measurement are not related to the study aim. 

3. In the results section, p-values for the main regression analysis should also be included for each analysis in the section. Without the p-values, we cannot know whether the relationship between the variables is significant or not.

4. In the discussion, the second and the third paragraphs are irrelevant to the study aim. These paragraphs deviate the readers from the main focus of the study.  

Reviewer 3 Report

The paper Association of Self-Reported Depression Symptoms with Physical Activity Levels in  Czechia concerns searching for associations between depressive disorder and undertaking various forms of physical activity in adults. In face of increasing number of depressive patients around the world, the researched issue is socially very important. The aim of the study was to evaluate, in a probability based-sample of the Czech population, the association between PA levels and different aspects of depression. The presented study called the Kardiovize study is a cross- sectional, randomized evaluation of adults between 25-64. The participants were 2 123 inhabitants  of Brno,  Czechia. The used measurement tools were : the Patient Health Questionnaire (PHQ-9),( Depressive symptoms) and the International Questionnaire of Physical Activity (IPAQ) long version (Self-reported PA).

Demographic data were gathered and laboratory analyses were performed . As statistical method the multivariable-adjusted Poisson regression models were carried out .The manuscript is generally clear, relevant for the field and presented in a well-structured manner. Cited references are current . Ethical  standards preserved. The strengths of the paper:

1.     Authors state depression is one of the significant determinants of disability-adjusted life years. The explored topic has not only scientific value but also practical one, as direction for promotion and prevention in the public health and psychiatry field.  multidimensional , with the strong  social, psychological and medical  background.

2.     The findings are of importance: the present study showed that PA levels have an inverse relationship with depressive symptoms, minimum a moderate level of PA is recommended for people with self-reported depression.

3.     The research results are evidence-based directions for public health policies .

4.     The very strong and worthy part of the paper is  discussion, which presents rich knowledge, deep interpretation and synthesis.

The weaknesses of the paper: There  are some mental- or information abbreviations. A few data are not precise enough and make authors’ consideration difficult to follow. The reception of the paper content will be more friendly with some additional, short explanations.

1.     Line 77

The study design was described previously [31].

This sentence is not friendly for the reader. It is worth  underlying the research design was published   (according to bibliographical data: McHugh RK, Weiss RD. Alcohol Use Disorder and Depressive Disorders. Alcohol Res. 2019;40(1):arcr.v40.1.01. doi:  392 10.35946/arcr.v40.1.01. PubMed PMID: 31649834.) but published not by authors of this paper .Of course – there is common tradition that researchers use methods and study plans developed by others. When presenting research design it will be appreciated to make reader familiar with study design. Perhaps this writing style is connected with language issues

2.   line 83

A random stratified sample by age and gender of 3,300 persons was adjusted for a  83 response rate of 64.4% (as projected from the Czech post-MONICA study).

It is not clear what MONICA-study is.

3.     line 106

Subjects classified as “high PA” were those who participated in a vigorous-intensity activity at least 3 days per week, achieving a minimum of 1500 MET-minutes/week,

 It will be nice to  be given the explanation, what MET- minutes mean.

Reviewer 4 Report

INTRODUCTION

I find it clear, well written and interesting.

It is missing the hypotheses of the study; I suggest including it together with the objective of the study (line 70).

MATERIALS AND METHOD

Adequate description of materials.

Accurate methodological design.

RESULTS

I suggest adding graphs to complement the presentation of the results.

DISCUSSION

I consider it is appropriate and well written. It highlights the main findings of the study.

The paper titled “Association of Self-Reported Depression Symptoms with Phys-ical Activity Levels in Czechia” addresses an interesting topic that provides evidence about the benefit of physical activity in the emotional state. In my opinion, these data are relevant for the promotion and prevention of mental health.
